# Replicable Reinforcement Learning

**Eric Eaton**
University of Pennsylvania
Philadelphia, PA 19104
eeaton@seas.upenn.edu

**Marcel Hussing**
University of Pennsylvania
Philadelphia, PA 19104
mhussing@seas.upenn.edu

**Michael Kearns**
University of Pennsylvania
Philadelphia, PA 19104
mkearns@cis.upenn.edu

**Jessica Sorrell**
University of Pennsylvania
Philadelphia, PA 19104
jsorrell@seas.upenn.edu

## Abstract

The *replicability crisis* in the social, behavioral, and data sciences has led to the formulation of algorithm frameworks for replicability — i.e., a requirement that an algorithm produce identical outputs (with high probability) when run on two different samples from the same underlying distribution. While still in its infancy, provably replicable algorithms have been developed for many fundamental tasks in machine learning and statistics, including statistical query learning, the heavy hitters problem, and distribution testing. In this work we initiate the study of *replicable reinforcement learning*, providing a provably replicable algorithm for parallel value iteration, and a provably replicable version of R-max in the episodic setting. These are the first formal replicability results for control problems, which present different challenges for replication than batch learning settings.

## 1 Introduction

The growing prominence of machine learning (ML) and its widespread adoption across industries underscore the need for replicable research [Wagstaff, 2012, Pineau et al., 2021]. Many scientific fields have suffered from this same inability to reproduce the results of published studies [Begley and Ellis, 2012]. Replicability in ML requires not only the ability to reproduce published results [Wagstaff, 2012], as may be partially addressed by sharing code and data [Stodden et al., 2014], but also consistency in the results obtained from successive deployments of an ML algorithm in the same environment. However, the inherent variability and randomness present in ML pose challenges to achieving replicability, as these factors may cause significant variations in results.

Building upon foundations of algorithmic stability [Bousquet and Elisseeff, 2002], recent work in learning theory has established rigorous definitions for the study of supervised learning [Impagliazzo et al., 2022] and bandit algorithms [Esfandiari et al., 2023a] that are provably *replicable*, meaning that algorithms produce identical outputs (with high probability) when executed on distinct data samples from the same underlying distribution. However, these results have not been extended to the study of control problems such as reinforcement learning (RL), that have long been known to suffer from stability issues [White and Eldeib, 1994, Mannor et al., 2004, Islam et al., 2017, Henderson et al., 2018]. These stability issues have already sparked research into robustness for control problems including RL [Khalil et al., 1996, Nilim and Ghaoui, 2005, Iyengar, 2005]. Non-deterministic environments and evaluation benchmarks, the randomness of the exploration process, and the sequential interaction of an RL agent with the environment all complicate the ability to make RL replicable. Our work is orthogonal to that of the robustness literature and our goal is not to reduce

37th Conference on Neural Information Processing Systems (NeurIPS 2023).

the effect of these inherent characteristics, such as by decreasing the amount of exploration that an agent performs, but to develop replicable RL algorithms that support these characteristics.

Toward this goal, we initiate the study of replicable RL and develop the first set of RL algorithms that are provably replicable. We contend that the fundamental theoretical study of replicability in RL might advance our understanding of the aspects of RL algorithms that make replicability hard. In this work, we put on a similar lens as Impagliazzo et al. [2022] and consider replicability as an algorithmic property that can be achieved simultaneously with exploration and exploitation. First, we show that it is possible to obtain a near-optimal, replicable policy given sufficiently many samples from every state in the environment. This notion is then naturally extended to replicable exploration.

Our contributions can be summarized as follows. We provide two novel and efficient algorithms to
- show that stochastic, sample-based value iteration can be done replicably and
- replicably explore the space of an MDP while also finding an optimal policy.

We experimentally validate that our algorithms require much fewer samples than theory suggests.

## 2 Preliminaries

### 2.1 Reinforcement learning

We consider the problem of solving a discounted Markov decision process (MDP) $\mathcal{M} = \{\mathcal{S}, \mathcal{A}, R, P, \gamma, \mu\}$ with state space $\mathcal{S}$, action space $\mathcal{A}$, reward function $R$, transition kernel $P$, discount factor $\gamma$, and initial state distribution $\mu$. We assume that the size of the state space $|\mathcal{S}|$ and number of possible actions $|\mathcal{A}|$ are finite and not too large. Further, we assume that the rewards for every state-action pair are deterministic, bounded, and known. Relaxing assumptions on the reward function might not necessarily seem straightforward in our goal of replicable RL, as the stochastic reward would need to be made replicable. However, the case can be handled by our algorithms with minor modifications and only constant factor overhead. The goal is to find a policy $\pi : \mathcal{S} \mapsto \mathcal{A}$ that maximizes the cumulative discounted reward $J_h = \sum_{k=h}^{\infty} \gamma^{k-h} R_k(s, a)$. We use the typical definitions of the value and Q-value functions for the expected cumulative discounted return from a state or state-action pair, respectively:

$$V_\pi(s) = \mathop{\mathbb{E}}_{\pi, P}[J_h | s_h = s] \qquad\qquad Q_\pi(s, a) = \mathop{\mathbb{E}}_{\pi, P}[J_h | s_h = s, a_h = a] \ .$$

To show the various difficulties that come from trying to achieve replicability in RL, we consider two different settings to examine various components of the problem.

**Parallel sampling setting** First, we ask whether it is even possible to obtain a replicable policy from empirical samples without considering the challenges of exploration. For this, we can adopt the setting of generative models $G_\mathcal{M}$, or more precisely, the parallel sampling setting. In the parallel sampling model, first introduced by Kearns and Singh [1998a], one has access to a parallel generative sampling subroutine $\mathbf{PS}(G_\mathcal{M})$. A single call to $\mathbf{PS}(G_\mathcal{M})$ will return, for every state-action pair $(s, a) \in \mathcal{S} \times \mathcal{A}$, a randomly sampled next state $s' \in \mathcal{S}$ drawn from $P(s'|s, a)$. The key advantage is that this model separates learning from the quality of the exploration procedure.

**Definition 2.1** (Generative model). *Let $\mathcal{M}$ denote an arbitrary MDP. Then a generative model $G_\mathcal{M}((s, a))$ is a randomized algorithm that, given a state-action pair $(s, a) \in \mathcal{S} \times \mathcal{A}$, outputs a deterministic reward $R(s, a)$ and a next state $s'$ sampled from $P(\cdot|s, a)$.*

**Definition 2.2** (Parallel sampling). *Let $\mathcal{M}$ denote an arbitrary MDP. Then a call to the parallel sampling subroutine $\mathbf{PS}(G_\mathcal{M})$ returns exactly one sample $s'_i \sim G_\mathcal{M}((s_i, a_i))$ for every state-action pair $(s_i, a_i)$ in $\mathcal{S} \times \mathcal{A}$ of $\mathcal{M}$ using a generative model.*

**Episodic setting** The second setting we consider is one in which an algorithm does have to explore the MDP before it can obtain an optimal policy. More precisely, we consider an episodic setting where, in every episode $e \in \{1, 2, ..., E\}$, the agent starts in a position $s_0 \sim \mu$ and interacts with the environment for a fixed amount of time $H$. At any step $h \in [1, H]$, the agent is in some state $s_h$, selects an action $a_h$, receives a reward $r_h$ and transitions to a new state $s_{h+1}$. Gathering a trajectory $\tau = (s_0, a_0, r_0, .., s_H, a_H, r_H)$ of states, actions and rewards under policy $\pi$ can be thought of as a draw from a distribution $\tau \sim P_\mathcal{M}^\pi(\tau)$. We will omit the sub-and superscripts when clear from context. For consistency with the remaining analysis, we work with a $\gamma$-discounted version of the problem.

## 2.2 Replicability

We build on the recent framework by Impagliazzo et al. [2022], which considers replicability as a property of randomized algorithms that take as input a dataset sampled i.i.d. from an arbitrary distribution. They consider an algorithm to be replicable if, on two runs in which its internal randomness is fixed and its input data is resampled, it outputs the same result with high probability:

**Definition 2.3** (Replicability). *Fix a domain $\mathcal{X}$ and target replicability parameter $\rho \in (0, 1)$. A randomized algorithm $\mathcal{A} : \mathcal{X}^n \to \mathcal{Y}$ is $\rho$-replicable if for all distributions $D$ over $\mathcal{X}$, randomizing over the internal randomness $r$ of $\mathcal{A}$ and choice of samples $S_1, S_2$, each of size $n$ drawn i.i.d. from $D$, we have:* $\boldsymbol{Pr}_{S_1, S_2, r}[\mathcal{A}(S_1; r) \neq \mathcal{A}(S_2; r)] \leq \rho$ .

Several key tools that were introduced by Impagliazzo et al. [2022] will prove useful or yield inspiration for the algorithms developed in this work. One of the key observations is that many of the computations in RL can be phrased as statistical queries, defined as follows:

**Definition 2.4** (Statistical query, [Kearns, 1998]). *Fix a distribution $D$ over $\mathcal{X}$ and an accuracy parameter $\alpha \in (0, 1)$. A statistical query is a function $\phi : \mathcal{X} \to [0, 1]$, and a mechanism $M$ answers $\phi$ with tolerance $\alpha$ on distribution $D$ if $a \leftarrow M$ satisfies $a \in [\mathbb{E}_{x \sim D}[\phi(x)] \pm \alpha]$.*

We will make direct use of the replicable algorithm for answering statistical queries by Impagliazzo et al. [2022] which will be useful to obtain replicable estimates of various measurements such as transition probabilities. We will refer to the replicable statistical query procedure as rSTAT. We note that Impagliazzo et al. [2022] also proves a lower-bound on the sample complexity required for replicable statistical queries, showing that the results below are essentially tight.

**Theorem 2.1** (Replicable statistical queries, Impagliazzo et al. [2022]). *There is a $\rho$-replicable algorithm rSTAT such that for any distribution $D$ over $\mathcal{X}$, replicability parameter $\rho \in (0, 1)$, accuracy parameter $\alpha \in (0, 1)$, failure parameter $\delta \in O(\rho)$, and query $\phi : \mathcal{X} \to [0, 1]$, letting $S$ be a sample of $n \in O\left(\frac{\log(1/\delta)}{(\rho - 2\delta)^2 \alpha^2}\right)$ elements drawn i.i.d. from $D$, we have that $a \leftarrow \mathsf{rSTAT}_{\alpha, \rho}(S, \phi)$ satisfies $a \in [\mathbb{E}_{x \sim D}[\phi(x)] \pm \alpha]$ except with probability at most $\delta$ over the samples $S$.*

At a very high level, rSTAT uses its sample to empirically estimate the expected value of the statistical query on the target distribution. It then uses its internal randomness to pick an evenly-spaced set of canonical representatives from the $[0, 1]$ interval, and returns whichever canonical representative is closest to the empirical estimate. We note that the algorithm of Impagliazzo et al. [2022] for replicably answering statistical queries is not only sample efficient, but also computationally efficient, as it has runtime polynomial in $1/\alpha$, $1/\rho$, and $\log(1/\delta)$.

## 3 Replicable reinforcement learning

To define replicability for the RL setting, we can adapt Definition 2.3 more or less exactly. The question that arises is which of the many RL objects should be made replicable? We separate the difficulty of replicability into three levels: replicability of the MDP, the value function, and the policy. Since these objects carry different amounts of information [Farahmand, 2011], the following relationships can be established.

If we are able to replicably (and accurately) estimate an MDP, we can always replicably compute an (optimal) value function using standard techniques on our estimates, and from replicable value functions we can obtain the corresponding policies. Note that the inverse is not true as we lose information when going from MDP to value function and then policy. As a result, we expect that replicable estimation of MDPs is the hardest setting in stochastic RL, followed by replicable value function, and then policy estimation.

For replicability of control problems, a sensible measure to ask for is the production of identical policies, which are the ultimate object of primary interest. We would at least like to ensure that with high probability, we can obtain identical optimal policies across two runs of our RL procedures:

**Definition 3.1** (Replicable policy estimation). *Let $\mathcal{A}$ be a policy estimation algorithm that outputs a policy $\widehat{\pi}^* : \mathcal{S} \mapsto \mathcal{A}$ given a set of trajectories $S$ sampled from an MDP. Algorithm $\mathcal{A}$ is $\rho$-replicable if, given independently sampled trajectory sets $S_1$ and $S_2$, and yielding policies $\widehat{\pi}_1^*$ and $\widehat{\pi}_2^*$, it holds*

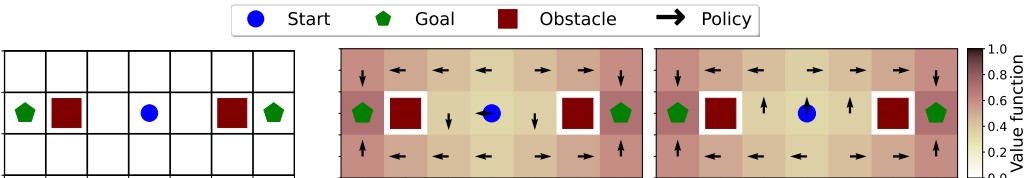

Figure 1: The GridWorld for our experiments (left) and two different policies that were generated by the Phased Q-learning Algorithm on this gridworld (center and right). Following the first policy (center) more likely reaches the left goal while following the right policy more likely reaches the right goal. All states except the goals have 0 reward. The actions are up, down, left and right; there is a 30% chance that after choosing an action the agent moves left or right of the target direction.

*for all states $s \in \mathcal{S}$ and actions $a \in \mathcal{A}$ that*

$$\boldsymbol{Pr}_{S_1, S_2, r}[\widehat{\pi}^{*(1)}(a|s) \neq \widehat{\pi}^{*(2)}(a|s)] \leq \rho$$
$$s.t. \ \widehat{\pi}^{*(1)}(a|s) \leftarrow \mathcal{A}(S_1; r) \quad \wedge \quad \widehat{\pi}^{*(2)}(a|s) \leftarrow \mathcal{A}(S_2; r) \ ,$$

*where $r$ represents the internal randomness of $\mathcal{A}$. Trajectory sets $S_1$ and $S_2$ may potentially be gathered from the environment during the execution of an RL algorithm.*

While this definition is the weakest we would like to achieve, the results we present in this paper provide stronger guarantees. Our Replicable Phased Value Iteration builds on [Kearns and Singh, 1998a] and ensures replicability of value functions, while our Replicable Episodic R-max follows [Kearns and Singh, 1998b, Brafman and Tennenholtz, 2003] and provides replicability of full MDPs. Equivalent formal definitions for replicable value and MDP estimation are given in Appendix A.

Current algorithms for sample-based RL problems will struggle to satisfy Definition 3.1 of replicability and output different policies even in simple environments (see Figure 1). In some cases, this may not be problematic since the resulting policies will still be $\varepsilon$-optimal, but in practice it is often hard to tell when that is the case. Fixing replicability will support the identification of problematic solutions and encourage procedures that yield more stable solutions in the long run. Varying policies can, for example, arise from sample uncertainty, insufficient state-space coverage, or differing exploration. In order to achieve replicability, all of the aforementioned challenges need to be addressed, which makes for an intricate but interesting problem. With this in mind, the next section will introduce a first set of formally replicable algorithms that separate out some of these challenges.

## 4 Algorithms

### 4.1 Replicable phased value iteration

The first question we answer positively is whether it is even possible to achieve replicability when the samples are drawn i.i.d. from the same distribution. For this, we use the parallel sampling model described in section 2. This model is well-suited for the task as it allows us to analyze sample-based value iteration independent of the exploration policy that collects the samples.

We provide a replicable version of indirect Phased Q-learning [Kearns and Singh, 1998a], which was later also referred to as Phased Value Iteration [Kakade, 2003]. In brief, the algorithm iterates $T$ times and at every iteration makes $m$ calls to $\mathbf{PS}(G_{\mathcal{M}})$, computes an approximate value estimate for every state and does one round of value updates. Kearns and Singh [1998a] provide the following Lemma 4.1 to show the optimality of the original procedure.

**Lemma 4.1** (Phased Q-learning convergence, [Kearns and Singh, 1998a]). *Suppose the number of calls to $\mathbf{PS}(G_{\mathcal{M}})$ is chosen such that the value function estimates produced in every round by Phased Q-learning are sufficiently accurate. For any MDP $\mathcal{M}$, Phased Q-learning converges to a policy $\widehat{\pi}^*$ whose return is within $\varepsilon$ of the optimal policy $\pi^*$.*

Our algorithm operates similarly, but we would like to achieve replicability on top of optimality. We use a randomized rounding procedure for statistical query estimation (rSTAT) provided by Impagliazzo et al. [2022] to compute the value estimates at every iteration. For this, we assume that the value function is normalized to the interval $[0, 1]$. A detailed description of our algorithm is

provided in Algorithm 1. The Replicable Phased Value Iteration (rPVI) algorithm we provide satisfies Definition 3.1 and produces $\varepsilon$-optimal policies. It goes even one step further and produces not only replicable policies but replicable value functions. This is formalized in the following Theorem 4.1.

**Theorem 4.1.** *Let $\varepsilon \in (0,1)$ be the accuracy and $\rho \in (0,1)$ be the replicability parameter. Let $\delta \in (0,1)$ be the sample failure probability. Set the number of calls to $\mathbf{PS}(G_{\mathcal{M}})$ at every iteration to*

$$m = O\left( \frac{\log^2(1/\varepsilon)|\mathcal{S}|^2|\mathcal{A}|^2}{\varepsilon^2(\rho - 2\delta)^2} \log\left( \frac{|\mathcal{S}||\mathcal{A}|}{\delta} + \log\log(1/\varepsilon) \right) \right)$$

*where $O$ supresses the dependence on $\gamma$. In two runs $(1)$ and $(2)$ with shared internal randomness, Algorithm 1 produces identical policies, s.t. $\mathbf{Pr}[\widehat{\pi}^{*(1)} \neq \widehat{\pi}^{*(2)}] \in O(\rho)$. In every run, the produced policies $\widehat{\pi}^*$ achieve return at most $\varepsilon$ less than the optimal policy $\pi^*$ with all but probability $O(\delta)$.*

*Proof Sketch.* We give a sketch for the proof of the theorem here and refer the reader to a full proof in Appendix B.2. Assume that we can get replicable and accurate estimates of the value function expectations from our rSTAT procedure. One can show by induction that the algorithm consistently produces the same value functions in every iteration. Lemma 4.2 guarantees the convergence to an optimal policy. Finally, we can use union and Chernoff bounds to pick a sufficiently large sample for our rSTAT queries to be replicable and accurate and satisfy our assumption.

An interesting observation is that rPVI discretizes the space of values as a function of the $\varepsilon$-parameter and $\gamma$ (see Appendix B.2). As a result, replicability becomes harder for larger values of $\gamma$ as discretization intervals become smaller and we require more samples to obtain an equally sized $\rho$. This is intuitive as we need to account for more potential future states that might impact our estimates.

The number of samples to compute a replicable value function is at most $O(\log^2(1/\epsilon)|\mathcal{S}|^2|\mathcal{A}|^2/\rho^2)$ times larger than computing a non-replicable one [Kearns and Singh, 1998a]. Still, a key observation of the original Phased Q-learning result was that it is sufficient for every state-action pair to have a sample size logarithmic in $|\mathcal{S}||\mathcal{A}|$, making the procedure cheaper than estimating the full transition dynamics of an MDP. The cost of replicability is the loss of this property. However, we note that rPVI does not yield replicable transition probability estimation. Using the idea of rSTAT queries to obtain transition estimates turns out to be significantly more expensive than the replicable value estimation done by Algorithm 1 (see Appendix B.2.1). Our results retain the notion that direct value estimation is much cheaper than estimating the full transition kernel even in the presence of replicability.

## 4.2 Replicable RL with exploration

Next, we consider the setting of episodic exploration. We show that, despite the stochastic nature of exploration, it is possible to guarantee replicability while still outputting an $\varepsilon$-optimal policy.

We take the R-max algorithm of Brafman and Tennenholtz [2003] as the starting point for our replicable algorithm RepRMAX (Algorithm 2). It proceeds in rounds where the agent interacts with

---

**Algorithm 1** Replicable Phased Value Iteration (rPVI)
Parameters: accuracy $\varepsilon$, failure probability $\delta$, replicability failure probability $\rho$
Input: Generative Model $G_{\mathcal{M}}$
Output: $\varepsilon$-optimal policy $\widehat{\pi}^*$

---

   Initialize $\widehat{Q}_0(s,a)$ to 0 for all $(s,a) \in \mathcal{S} \times \mathcal{A}$
   For all $s \in \mathcal{S}$, let $\phi_Q(s) := \max_a Q(s,a)$
   **for** $t = 0, \cdots, T-1$ **do**
      $S \leftarrow (\mathbf{PS}(G_{\mathcal{M}}))^m$              ▷ do $m$ calls to $\mathbf{PS}(G_{\mathcal{M}})$ and store next-states in a map
                                        from state-action pairs $(s,a)$ to next states $S[(s,a)]$.

      **for** $(s,a) \in \mathcal{S} \times \mathcal{A}$ **do**
         $\widehat{V}(s') \leftarrow \mathsf{rSTAT}(S[(s,a)], \phi_{\widehat{Q}_t}(s'))$
         $\widehat{Q}_{t+1}(s,a) \leftarrow R(s,a) + \gamma\widehat{V}(s')$
      **end for**
   **end for**
   **return** $\widehat{\pi}^* = \arg\max_a \widehat{Q}_T(s,a)$

---

---

**Algorithm 2** Replicable Episodic R-max (RepRMAX)

Parameters: Accuracy $\varepsilon$, accuracy failure probability $\delta$, replicability failure probability $\rho$, horizon $H$

Input: MDP $\mathcal{M}$, maximum reward $R_{\max}$

Output: $\varepsilon$-optimal policy $\pi_{\hat{\mathcal{M}}_K}$

---

Initialize $\pi_{\hat{\mathcal{M}}_K}$ to a random policy, counters for state-visitation $n(s, a)$ to 0

Initialize $K$, the set collecting known state-action pairs, to the empty set $\emptyset$

Initialize $S$, the set collecting trajectories to be used for estimating transition probabilities, to $\emptyset$

Initialize $\widehat{\mathcal{M}}_K$ as $\widehat{P}_K(s'|s, a) \coloneqq \mathbb{1}[s' = s]$ for all $(s, a, s')$ and $\widehat{R}_K(s, a) \coloneqq R_{\max}$ for all $(s, a)$

$i = 1$

**while** $\pi_{\hat{\mathcal{M}}_K}$ is not $\varepsilon$-optimal **do**

    Collect a sample of trajectories $S_i \leftarrow P(\tau)^m$ and add $S_i$ to $S$

    $K_i \leftarrow \mathsf{RepUpdateK}(S_i, K, \{n(s, a)\}_{(s,a) \in \mathcal{S} \times \mathcal{A}})$, identify new known states

    For all $(s, a) \in K_i$, let $S[(s, a)]$ be the multiset of $s'$ visited from $(s, a)$ for all $\tau \in S$

    For all $s' \in \mathcal{S}$, let $\phi_{s'}(s) \coloneqq \mathbb{1}[s = s']$

    Update $\widehat{\mathcal{M}}_K$ for all $(s, a) \in K_i$: $\widehat{P}_K(s'|s, a) \coloneqq \mathsf{rSTAT}(S[(s, a)], \phi_{s'})$, $\widehat{R}_K(s, a) \coloneqq R(s, a)$

    $K = K \cup K_i$

    Compute $\pi_{\hat{\mathcal{M}}_K}$ from $\widehat{\mathcal{M}}_K$

**end while**

**return** $\pi_{\hat{\mathcal{M}}_K}$

---

the environment for multiple episodes. The collection of trajectories encountered during exploration is used to incrementally build a model $\widehat{\mathcal{M}}$ of the underlying MDP $\mathcal{M}$. The algorithm implicitly partitions the set of state-action pairs $\mathcal{S} \times \mathcal{A}$ into two groups: known and unknown. All $(s, a) \in \mathcal{S} \times \mathcal{A}$ are initialized to be unknown. While a state is unknown, the model $\widehat{\mathcal{M}}$ maintains that $(s, a)$ is a self-loop with probability 1, and that $(s, a)$ has maximum reward, thereby promoting exploration of unknown states. After a state-action pair $(s, a)$ has been visited sufficiently many times, it is added to the collection of known states $K$ and its transition probabilities $\widehat{P}$ and reward $\widehat{R}$ are updated with an empirical approximation of $\widehat{P}_K(s' \mid s, a)$ for all $s' \in \mathcal{S}$ and the observed reward $R$, respectively. After every update, the policy $\pi_{\widehat{\mathcal{M}}_K}$ is computed as the optimal policy of the current model estimate.

While convergence of Algorithm 2 to an $\varepsilon$-optimal policy follows from familiar arguments [Brafman and Tennenholtz, 2003], proving replicability will require a great deal of additional care. To ensure that two runs of RepRMAX (with shared internal randomness) converge to the same policy with high probability, we will show something even stronger: we prove that two such runs will with high probability perform the same sequence of updates to their respective models $\widehat{\mathcal{M}}_K$ and policies $\pi_{\hat{\mathcal{M}}_K}$.

To enforce this property, we introduce a sub-routine in Algorithm 3 which replicably identifies state-action pairs that should be added to the collection of known states. Guaranteeing that at each iteration the set of known states $K$ will be the same for two independent runs of the algorithm helps ensure that the models of the MDP $\widehat{\mathcal{M}}_K$, and consequently the policies $\pi_{\hat{\mathcal{M}}_K}$ learned at each iteration, will also be identical. To provide replicability, we will want to avoid using a fixed threshold for the number of times a state-action pair $(s, a)$ must be visited before it is considered "known". Under small deviations in realized transitions, a fixed threshold might lead to some $(s, a)$ becoming known in one run of the algorithm and not another. Instead, we use a randomized threshold.

In a call to Algorithm 3, the sample drawn at that round is used to estimate the expected number of visits to $(s, a)$ in a single trajectory, for every $(s, a)$. This estimate is added to the count $n(s, a)$, which maintains the sum, over all iterations thus far, of the estimated expected visits to $(s, a)$ from a single trajectory of the policy $\pi_{\hat{\mathcal{M}}_K}$ at that iteration. A new threshold $k'$ is then sampled uniformly from $[k, k + w]$. If $n(s, a) \geq k'$, it is added to the set of known states $K$. From standard concentration arguments, we know that for two runs of Algorithm 2 with independent samples, the estimates of the total number of expected visits $n(s, a)$ will both be close to the true total number of expected visits, and therefore close to each other. Algorithm 3 will only make different decisions about adding an $(s, a)$ pair to the set of known states if the threshold $k'$ is chosen to fall between the two estimated values $n(s, a)$ from the two runs. Here, the concentration of $n(s, a)$ and the fact that $k'$ is randomized allows us to bound the probability that the threshold $k'$ is chosen to fall between the different $n(s, a)$

values. We show in Theorem 4.2 that so long as the sample size $m$ that is used to estimate expected visits, and the window $w$ from which the randomized threshold in sampled, are taken to be large enough, the update to the set of known states at each round will be replicable.

---

**Algorithm 3** RepUpdateK

Parameters: Accuracy failure probability $\delta$, replicability failure probability $\rho$
Input: Sample of trajectories $S_i$, set of known states $K$, set of state-visit counts $\{n(s,a)\}_{(s,a)\in \mathcal{S}\times \mathcal{A}}$
Output: List of new known state-action pairs $K_i$

---

$\quad K_i = \{(s,a) : (s,a) \in \mathcal{S} \times \mathcal{A} \text{ and } (s,a) \notin K\}$
$\quad k' \leftarrow \mathcal{U}[k, k+w]$
$\quad$ **for** $(s,a) \in K_i$ **do**
$\quad\quad \widehat{c}_{s,a} = \frac{1}{|S_i|} \sum_{\tau \in S_i} \sum_{h=1}^{H} \mathbb{1}[(s_h, a_h) = (s,a)]$
$\quad\quad n(s,a) = n(s,a) + \widehat{c}_{s,a}$
$\quad\quad$ **if** $n(s,a) < k'$ **then**
$\quad\quad\quad$ Remove $(s,a)$ from $K_i$
$\quad\quad$ **end if**
$\quad$ **end for**
$\quad$ **return** $K_i$

---

Now that we have understood the intricacies on an intuitive level, we will prove convergence (Lemma 4.2) and replicability (Theorem 4.2) of Algorithm 2.

**Lemma 4.2** (Convergence). *Consider $\mathcal{A}$ to be Algorithm 4. Let $\varepsilon \in (0, 1)$ be the accuracy parameter, $\rho \in (0, 1)$ the replicability parameter, and $\delta \in (0, 1)$, be the sample failure probability, with $\delta < \rho/4$. Let $T \in \Theta(\frac{H|\mathcal{S}||\mathcal{A}|}{\varepsilon} + \frac{H^2 \log(1/\delta)}{\varepsilon^2})$ be a bound on the number of iterations of Algorithm 2. Suppose $1 - \gamma > \frac{\sqrt{\varepsilon}}{H|\mathcal{A}|}$ and let $m \in \tilde{O}\left(\frac{|\mathcal{S}|^2|\mathcal{A}|^2 T^4 \log(1/\rho)}{\rho^2}\right)$ be the number of trajectories per iteration. Let $k = H$ be the lowest expected visit count of a state-action pair before it is known. Let $w \in O(k)$ define the window $[k, k+w]$ for sampling the randomized threshold $k'$. Then with all but probability $\delta$, after $T$ iterations, $\mathcal{A}$ yields an $\varepsilon$-optimal policy.*

The proof of convergence is similar to those found in [Kearns and Singh, 1998b] and [Brafman and Tennenholtz, 2003], so we defer the proof to Appendix B.3. We continue here with the final theorem statement that summarizes the properties of our RepRMAX algorithm.

**Theorem 4.2.** *Let parameters be set as in Lemma 4.2. Then with all but probability $\delta$, $\mathcal{A}$ converges to an $\varepsilon$-optimal policy in $T$ iterations and samples $mT$ trajectories, each of length $H$, for a total sample complexity of $O\left(\frac{|\mathcal{S}|^7|\mathcal{A}|^7 H^6 \log(1/\rho)}{\rho^2 \varepsilon^5} + \frac{|\mathcal{S}|^2|\mathcal{A}|^2 H^{10} \log^5(1/\delta) \log(1/\rho)}{\varepsilon^{10}}\right)$. Further, let $S_1$ and $S_2$ be two trajectory sets, independently sampled over two runs of $\mathcal{A}$ with shared internal randomness, and let $\pi_{\hat{\mathcal{M}}_K}^{(1)}(a|s) \leftarrow \mathcal{A}(S_1; r)$ and $\pi_{\hat{\mathcal{M}}_K}^{(2)}(a|s) \leftarrow \mathcal{A}(S_2; r)$. Then*

$$\boldsymbol{Pr}_{S_1, S_2, r}\left[\pi_{\hat{\mathcal{M}}_K}{}^{(1)}(a|s) \neq \pi_{\hat{\mathcal{M}}_K}{}^{(2)}(a|s)\right] \in O(\rho).$$

*Proof.* Lemma 4.2 gives us that, for our settings of $k$ and $T$, Algorithm 2 converges to an $\varepsilon$-optimal policy in $T$ iterations, except with probability $\delta$. The sample complexity follows immediately from the bound on $T$ and the setting of $m$, so it remains to analyze replicability. Our analysis will make use of some additional shorthand. We use $\rho_K \in O(\rho/(T|\mathcal{S}||\mathcal{A}|))$ to denote the replicability parameter for the decision to add a single $(s,a)$ to $K$, in a single call to Algorithm 3. We similarly use $\rho_{SQ} \in O(\rho/(|\mathcal{S}|^2|\mathcal{A}|))$, $\alpha_{SQ} \in O(\varepsilon(1-\gamma)^2/|\mathcal{S}|)$, and $\delta_{SQ} \in O(\delta/(|\mathcal{S}|^2|\mathcal{A}|))$ to denote the replicability, accuracy, and failure parameters for the rSTAT queries made during the updates to $P(s'|s,a)$. We use $t \in O(\frac{w\rho_K}{T}) \in O(\frac{H\rho}{|\mathcal{S}||\mathcal{A}|T^2})$ to denote a high probability bound on the difference between the empirical estimates for the expected visits to a given $(s,a)$ in a trajectory across two runs of Algorithm 3, i.e. $|\widehat{c}_{s,a}^{(1)} - \widehat{c}_{s,a}^{(2)}| \in O(t)$. We are now ready to prove the following stronger claim:

**Claim 4.1.** *If two runs of Algorithm 2 begin iteration $i$ with $\widehat{\mathcal{M}}_K^{(1)} = \widehat{\mathcal{M}}_K^{(2)}$, $\pi_{\hat{\mathcal{M}}_K}^{(1)} = \pi_{\hat{\mathcal{M}}_K}^{(2)}$, and $|n(s,a)^{(1)} - n(s,a)^{(2)}| \in O(it) \,\forall (s,a)$, then at the end of $i$, $\widehat{\mathcal{M}}_K^{(1)} = \widehat{\mathcal{M}}_K^{(2)}$, $\pi_{\hat{\mathcal{M}}_K}^{(1)} = \pi_{\hat{\mathcal{M}}_K}^{(2)}$, and $|n(s,a)^{(1)} - n(s,a)^{(2)}| \in O(it + t) \,\forall (s,a)$, with all but probability $O(\rho_K|\mathcal{S}||\mathcal{A}| + \rho_{SQ}|K_1||\mathcal{S}|)$.*

We take the initialization of Algorithm 2 as the base case for our inductive proof. Before the first iteration, $\pi_{\hat{\mathcal{M}}_K}$ is initialized randomly and shared internal randomness yields $\pi_{\hat{\mathcal{M}}_K}^{(1)} = \pi_{\hat{\mathcal{M}}_K}^{(2)}$. We deterministically initialize $\widehat{\mathcal{M}}_K$ and all $n(s,a)$, and so $\widehat{\mathcal{M}}_K^{(1)} = \widehat{\mathcal{M}}_K^{(2)}$ and $n(s,a)^{(1)} = n(s,a)^{(2)}$.

Next, we prove the inductive step. We begin by showing that, at the end of the $i$th iteration, $|n(s,a)^{(1)} - n(s,a)^{(2)}| \in O(it+t) \; \forall (s,a)$, with all but probability $O(\rho_K|\mathcal{S}||\mathcal{A}|)$.

Our inductive hypothesis gives us that $|n(s,a)^{(1)} - n(s,a)^{(2)}| \in O(it)$, so it suffices to show that, for a single $(s,a)$, $\widehat{c}_{s,a}^{(1)} - \widehat{c}_{s,a}^{(2)} \in O(t)$ except with probability $O(\rho_K)$. To obtain high probability bounds on $|\widehat{c}_{s,a}^{(1)} - \widehat{c}_{s,a}^{(2)}|$, we will rely on our assumption that at the start of the iteration, $\pi_{\hat{\mathcal{M}}_K}^{(1)} = \pi_{\hat{\mathcal{M}}_K}^{(2)}$. It follows that, for every state-action pair $(s,a)$, the expected number of visits to $(s,a)$ in a single episode is the same for both iterations. That is, for every $(s,a)$, defining $c_{s,a} := \mathbb{E}_{\tau \sim P(\tau)} \left[ \sum_{h=1}^H \mathbb{1}[(s_h,a_h) = (s,a)] \right]$, we have $c_{s,a}^{(1)} = c_{s,a}^{(2)}$.

For a particular $(s,a)$, Chernoff bounds applied to the average observed counts $\widehat{c}_{s,a}^{(1)}$ and $\widehat{c}_{s,a}^{(2)}$ show that they must both be close to their (shared) expectation with high probability. We draw a sample of

$$m \in O\left( \frac{|\mathcal{S}|^2|\mathcal{A}|^2 T^4 \log(1/\rho)}{\rho^2} \right) \in O\left( \frac{H^2 \log(1/\rho)}{t^2} \right) \in \tilde{O}\left( \frac{H^2 \log(1/\rho_K)}{t^2} \right)$$

trajectories, and each $c_{s,a} \in [0,H]$, so except with probability $4\exp\left( \frac{-2t^2 m^2}{H^2 m} \right) \in O(\rho_K)$,

$$|\widehat{c}_{s,a}^{(1)} - \widehat{c}_{s,a}^{(2)}| \leq \left| \frac{1}{m} \sum_{\tau \in S_1^{(1)}} \sum_{h=1}^H \mathbb{1}[\tau_h = (s,a)] - c_{s,a}^{(1)} \right| + \left| \frac{1}{m} \sum_{\tau \in S_1^{(2)}} \sum_{h=1}^H \mathbb{1}[\tau_h = (s,a)] - c_{s,a}^{(1)} \right| \in O(t),$$

where $\tau_h := (s_h, a_h)$. Union bounding over all $s \in \mathcal{S}$ and $a \in \mathcal{A}$ shows that the stated bound holds for all $(s,a)$ except with probability $\rho_K|\mathcal{S}||\mathcal{A}|$.

We now show that $\widehat{\mathcal{M}}_K^{(1)} = \widehat{\mathcal{M}}_K^{(2)}$ at the end of the iteration, except with probability $O(\rho_K|\mathcal{S}||\mathcal{A}| + \rho_{SQ}|K_i||\mathcal{S}|)$. Observe that $\widehat{\mathcal{M}}_K^{(1)} = \widehat{\mathcal{M}}_K^{(2)}$ at the end of the iteration unless at least one of the following two events occurs: 1) $K_1^{(1)} \neq K_1^{(2)}$ - the set of new known $(s,a)$ pairs differs across the two runs. 2) The updates to $\widehat{P}_K(s'|s,a)$ and $R(s,a)$ differ for at least one $(s,a)$. The first event occurs exactly when $k'$ falls in between $n(s,a)^{(1)} + \widehat{c}_{s,a}^{(1)}$ and $n(s,a)^{(2)} + \widehat{c}_{s,a}^{(2)}$. We have already shown that

$$|n(s,a)^{(1)} + \widehat{c}_{s,a}^{(1)} - n(s,a)^{(2)} - \widehat{c}_{s,a}^{(2)}| \in O(it+t) \in O(tT),$$

for a single $(s,a)$, except with probability $O(\rho_K)$. We have sampled $k'$ uniformly at random from an interval of width $w$, so it follows that $\mathbf{Pr}_{k',S_1,S_2}[(s,a) \in K_i^{(1)} \triangle K_i^{(2)}] \in O(\rho_K + tT/w)$. We took $t \in \frac{w\rho_K}{T}$, so by union bound over $\mathcal{S} \times \mathcal{A}$, the probability of the first event is at most $O(|\mathcal{S}||\mathcal{A}|\rho_K)$.

To bound the probability of the second event conditioned on the first event not occurring, it suffices to bound the probability that the updates to $\widehat{P}_K(s'|s,a)$ for $(s,a) \in K_i$ differ across both runs, By the conditioning, we have $K_1^{(1)} = K_1^{(2)}$, so it suffices to show that each call to rSTAT returns the same value for both runs. Taking $\rho_{SQ}$, $\alpha_{SQ}$, and $\delta_{SQ}$ as the replicability, tolerance, and failure parameters respectively gives that a sample of size $s \in O(|\mathcal{S}|^2 \log(1/\delta_{SQ})/((\varepsilon(\rho_{SQ} - 2\delta_{SQ}))^2(1-\gamma)^4)$ is sufficient, by Theorem 2.1. Furthermore, we have assumed that $1 - \gamma > \frac{\sqrt{\varepsilon}\log^{1/4}(1/\delta)}{H|\mathcal{A}|\log^{1/4}(1/\rho)}$, $\delta_{SQ} < \rho_{SQ}/4$, and $\rho_{SQ} \in O(\rho/|\mathcal{S}|^2|\mathcal{A}|)$, so a sample of size $s \in O\left( \frac{|\mathcal{S}|^6|\mathcal{A}|^6 H^4 \log(1/\rho)}{\varepsilon^4 \rho^2} \right)$ will also suffice. Each $(s,a)$ is added to $K_i$ only if it was visited at least $km$ times. We have taken $k = H$, $m \in O\left( \frac{|\mathcal{S}|^2|\mathcal{A}|^2 T^4 \log(1/\rho)}{\rho} \right)$, and $T \in \Omega\left( \frac{|\mathcal{S}||\mathcal{A}|H}{\varepsilon} \right)$. It follows that $mk \in O\left( \frac{|\mathcal{S}|^6|\mathcal{A}|^6 H^5 \log(1/\rho)}{\varepsilon^4 \rho^2} \right)$ and therefore $S[(s,a)]$ comprises at least $s$ i.i.d. samples from $P(\cdot \mid s,a)$, as desired. Union bounding over the $|K_i||\mathcal{S}|$ queries in the $i$th iteration gives a bound of $|K_i||\mathcal{S}|\rho_{SQ}$ on the probability of the second event, conditioned on the first event not happening.

We now assemble our inductive argument into a proof of the theorem. At the start of iteration $i$, the inductive hypothesis holds except with probability $\sum_{j=1}^{i-1} \rho_K|\mathcal{S}||\mathcal{A}| + \rho_{SQ}|K_j||\mathcal{S}|$.

Noting that $\sum_{j=1}^{T} |K_j| \leq |\mathcal{S}||\mathcal{A}|$, and recalling that we have taken replicability parameters $\rho_K \in O(\rho/(T|\mathcal{S}||\mathcal{A}|))$ and $\rho_{SQ} \in O(\rho/(|\mathcal{S}|^2|\mathcal{A}|))$, ensures we achieve a replicability parameter $\rho$ after the $T$ iterations of Algorithm 2. $\qquad\qquad\qquad\qquad\qquad\qquad\qquad\square$

### 4.3 Limitations

As mentioned previously, our bounds lose some of the properties that standard RL results provide, such as the ability to estimate value functions with only a logarithmic dependence on relevant parameters. We expect that some of the sample complexity overhead from achieving replicability is inevitable, as seen in the statistical query lower-bound of Impagliazzo et al. [2022]. Nonetheless, we hope that future work can improve on the sample-complexities of our algorithms.

Our work is in part motivated by the recent replicability concerns in deep RL [Islam et al., 2017, Henderson et al., 2018]. However, establishing formal guarantees in these highly complicated settings is often not easy. As such, our algorithms suffer the weakness that many theoretical results in RL have to deal with, namely their lack of immediate applicability to real-world problems. Yet, our empirical evaluation in section 5 will show that there is hope for practical application.

## 5   Experiments

While our asymptotic bounds have sample complexity overhead from the introduction of replicability, we would like to analyze the actual requirements in practice. We introduce a simple MDP in Figure 1 that contains several ways of reaching the two goals. We analyze the impact of the number of calls to $\mathbf{PS}(G_{\mathcal{M}})$ on replicability for rPVI. In theory, our dependence on the number of calls is not logarithmic with respect to $|\mathcal{S}||\mathcal{A}|$ but we would like to see if can draw a sample that is much smaller, maybe even on the order of the logarithmic requirement. We choose accuracy $\varepsilon = 0.02$, failure rate $\delta = 0.001$ and replicability $\rho = 0.2$. The number of calls that would be required by standard Phased Q-learning is at most $m \approx 13000$ (ignoring $\gamma$ factors). We take several multiples of $m$ and measure the fraction of identical and unique value functions, treating the rSTAT $\rho_{SQ}$ as a hyperparameter.

The results are presented Figure 2, revealing that the number of samples needed to replicably produce the same value function can be several orders of magnitude lower than suggested by our bounds and that it is feasible to use a larger $\rho_{SQ}$ than theoretically required. This should allow us to scale to more complex problems in the future. The algorithm quickly produces a small set of value functions that may not be identical but, with a little more data, minor differences are removed. Note that using a replicable procedure naturally incurs overhead, which is expected. However, the overhead is significantly better than the theoretically required sample-size with squared $|\mathcal{S}||\mathcal{A}|$ dependence. In the rSTAT procedure, taking smaller values for $\rho_{SQ}$ for a fixed sample should improve replicability at the cost of accuracy of query responses, by increasing the width of each subinterval of the partition so that there are fewer partition elements overall. The experiments highlight that, as long as sample sizes are sufficiently large and $\rho_{SQ}$ is chosen small enough, we achieve high replicability.

## 6   Related work

Our work builds upon the foundational ideas by Impagliazzo et al. [2022], who introduce formal notions of replicability that are strongly related to robustness, privacy, and generalization [Bun et al., 2023, Kalavasis et al., 2023]. Building on these formal definitions of replicability, researchers have provided algorithms for replicable bandits [Esfandiari et al., 2023a] and replicable clustering [Esfandiari et al., 2023b]. Ahn et al. [2022] introduce algorithms for convex optimization using a slightly different notion of replicability. Our paper presents the first results for formally replicable algorithms in a control setting.

A concurrent and independent work by Karbasi et al. [2023] also studies formal replicability of reinforcement learning. They also study the setting of discounted tabular MDPs, with access to a generative model, and show the same sample complexity upper-bounds for achieving replicable policy estimation in this setting that we prove in our work. Additionally, they provide a matching lower bound. They go on to consider two relaxed notions of replicability that allow them to provide improved sample complexity upper-bounds in the generative model setting. Our work instead considers a second setting, providing a first algorithm for replicable policy estimation in the episodic exploration

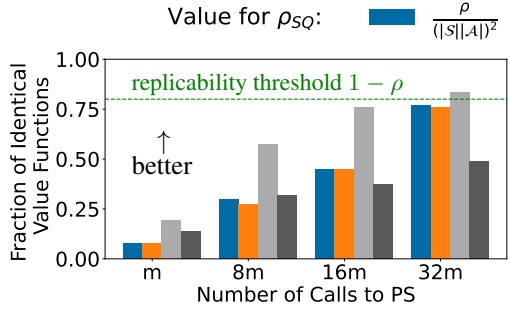
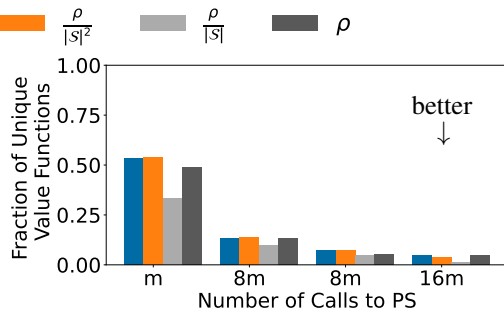

The largest percentage of identical value functions across 150 runs. With more data, the quantity increases and the choice of $\rho_{SQ}$ becomes less important.

The percentage of unique value functions across 150 runs. Varying $\rho_{SQ}$ has negligible impact, while more samples quickly reduce it.

Figure 2: The rPVI algorithm evaluated on varying numbers of calls to $\mathbf{PS}(G_{\mathcal{M}})$, with several values for the internal rSTAT parameter $\rho_{SQ}$. Results are provided across 150 runs with different random sampling seeds. The number of calls is set to constant factor multiples of $m = 13000$. The dotted green line denotes the replicability threshold of $1 - \rho$. The results show that, in practice, the number of samples needed for replicability can be orders of magnitude lower than our bounds suggest.

setting. We also provide experimental validation of the practical feasibility of our Replicable Phased Value Iteration algorithm.

From an RL perspective, our work is strongly related to understanding exploration in MDPs [Kearns and Singh, 1998b, Brafman and Tennenholtz, 2003, Kakade, 2003]. In the finite-horizon episodic setting, researchers made progress on upper bounds for exploration Auer and Ortner [2006], Auer et al. [2008], Jaksch et al. [2010] that ultimately led to the development of a near-complete understanding of the problem [Azar et al., 2017, Zanette and Brunskill, 2019, Simchowitz and Jamieson, 2019]. Lower bounds are provided in other works [Dann and Brunskill, 2015, Osband and Roy, 2016]. Further, Jin et al. [2020], Kaufmann et al. [2021] provide results on a reward-free framework that allows for the optimization of any reward function. While a good amount of progress has been made on understanding the base problem, the notion of replicability is not considered in any of them.

Given the connections of replicability and robustness, our work is related but orthogonal to that of the study of worst-case optimal policies and value functions. These worst-case results are often obtained via the study of robust Markov decision processes, first introduced by Nilim and Ghaoui [2005], Iyengar [2005]. One line of work here has focused on relaxation of assumptions and combatting conservativeness in robust MDPs [Wiesemann et al., 2013, Mannor et al., 2016, Petrik and Russel, 2019, Panaganti and Kalathil, 2022]. Others have focused on various new formulations such as distributional robustness [Xu and Mannor, 2010, Yu and Xu, 2016]. However, all of the above work focuses on understanding worst-cases and finding policies that do not have to be replicable.

Finally, our work is related to efforts in practical RL to ensure replicability, such as benchmark design [Guss et al., 2021, Mendez et al., 2022] and robust implementation [Nagarajan et al., 2018, Seno and Imai, 2022] and evaluation [Lynnerup et al., 2020, Jordan et al., 2020, Agarwal et al., 2021].

## 7 Conclusion & future work

We introduced the notion of formal replicability to the field of RL and established various novel algorithms for replicable RL. While these first results might have sub-optimal sample complexities, they highlight the crucial fact that replicability in RL is hard and requires study of the various aspects that impact it. We hope that future work can alleviate some of these efficiency challenges. A general open question is if replicable RL might simply be harder by nature than standard RL? This question needs to be posed on various levels because, as we argue in Section 3, finding a replicable policy might be easier than requiring the value function to be replicable. Finally, we believe the development of replicable algorithms for other settings such as the non-episodic setting as well as practical application are of great importance.

## Acknowledgments and Disclosure of Funding

The first and second authors are partially supported by the DARPA SAIL-ON program under contract HR001120C0040, the DARPA ShELL program under agreement HR00112190133, and the Army Research Office under MURI grant W911NF20-1-0080. The last author is supported in part by the Simons Collaboration on the Theory of Algorithmic Fairness, and NSF grant CCF-2217062.

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
