# OpenReview forum: "Replicable Reinforcement Learning"
_NeurIPS.cc/2023/Conference — NeurIPS 2023 poster_

### Official Review · Reviewer_vrU3 · 2023-07-04

**Soundness:** 3 good
**Presentation:** 3 good
**Contribution:** 3 good
**Rating:** 7
**Confidence:** 2

**Summary:**


This paper discusses the development of algorithm frameworks for replicability, which is a response to the replicability crisis in social, behavioral, and data sciences. The paper introduces provably replicable algorithms for machine learning and statistics, including replication results for control problems, which pose different challenges than batch learning settings. The paper provides a provably replicable algorithm for parallel value iteration and a provably replicable version of R-Max in the episodic setting, which are the first formal replicability results for control problems.

**Strengths:**

1. The proposed method has a strong mathematical and theoretical analysis, which is convincing.
2. The researched field appears to be valuable and interesting.
3. The writing is clear and easy to understand.

**Weaknesses:**

Although the method has a strong mathematical and theoretical analysis, it would be better to have more sufficient experiments in the experimental section.

**Questions:**

1. Can the corresponding analysis continue to be maintained in the setting of deep networks?
2. How much additional cost is required for the proposed method?

**Limitations:**

Although the method has a strong mathematical and theoretical analysis, it would be better to have more sufficient experiments in the experimental section.

---

> ### Author Rebuttal · Authors · 2023-08-08
>
> Dear Reviewer vrU3,
>
> Thank you for your feedback. We are glad that you find our work convincing and valuable.
>
> In response to the comments in the Weaknesses and Questions sections:
>
> “Although the method has a strong mathematical and theoretical analysis, it would be better to have more sufficient experiments in the experimental section.”
> * Our algorithms are tabular reinforcement learning algorithms and we evaluate our algorithms on a tabular MDP in section 5. The experiments we conduct are explicitly designed with two goals in mind: a proof of concept for replicable algorithms in a non-trivial environment (i.e., an environment where there is more than one optimal policy, so that replicability is not already implied by optimality) and to show that the sample complexity overhead of replicability is not as onerous as might be suggested from the asymptotic upper-bounds. Our experiments cover both points. That being said, if there are suggestions for experiments that might improve our understanding of replicability in practice, we would be happy to run them!
>
> “Can the corresponding analysis continue to be maintained in the setting of deep networks?”
> * The proposed techniques are in their current form not applicable to non-linear function approximation.  Even linear function approximation would require the development of completely new tools for replicability. We agree that this is an interesting direction for future work but there are several leaps to make before we can guarantee replicability of deep learning approaches.
>
> "How much additional cost is required for the proposed method?”
> * The overhead on the methods is on the order $|S|^2|A|^2 / \rho^2$ for rPVI (see L173) and $|S|^5|A|^6H^6 /(\epsilon^2 \rho^2)$ for RepRMax (we will add this to the appendix)

---

> > ### Comment · Reviewer_vrU3 · 2023-08-15
> > **Keep Score**
> >
> > I still find this paper interesting at the moment, so I keep my score.

---

### Official Review · Reviewer_u4VH · 2023-07-05

**Soundness:** 3 good
**Presentation:** 3 good
**Contribution:** 2 fair
**Rating:** 6
**Confidence:** 3

**Summary:**

This paper studies an important topic of RL replicability. Under some assumptions, this work gives definitions of rho-replicable and proposes two algorithms: Rep-PVI and Rep-RMAX, and shows their reproducibility properties through proof.


**Strengths:**

Overall, the development of the paper is smooth, the topic is important, the perspective is novel in the RL literature, and the analytical and empirical evidence seems supportive of the claims.


**Weaknesses:**

Please see the questions section below


**Questions:**

Finite state space and known deterministic reward function seem to be too strong assumptions. The stochasticity comes from 1. exploration and 2. environment stochasticity (including dynamics and reward). If the reward is known and deterministic, a planning algorithm can always be applied and RL is not necessary.

With the parallel sampling definition, the parallel sampling subroutine PS(G) actually turns the transition distribution into a deterministic kernel, this is also a strong assumption. To this end, all stochasticity sources are canceled out in your study.

I may understand it wrongly, but with those assumptions, I wonder if the basic Q-learning algorithm also enjoys the property of convergence to the optimal policy with probability 1. This lead to another question of why the PVI and RMAX algorithms are selected to build the method upon.

For the experiment section, have the authors tried to experiment on some standard RL environment? Inclusively the classical navigation tasks (but are more standard ones) so that the trade-offs between replicability and the performance can have a clearer exposition?



**Limitations:**

The generality of the proposed method, which also affects the importance of their contribution to the field.

---

> ### Author Rebuttal · Authors · 2023-08-08
>
> Dear Reviewer u4VH,
>
> Thank you for your feedback! We are glad that you find the topic of replicable RL important and novel.
>
> In response to the comments in the Questions section:
>
>  “I may understand it wrongly, but with those assumptions, I wonder if the basic Q-learning algorithm also enjoys the property of convergence to the optimal policy with probability 1. This lead to another question of why the PVI and RMAX algorithms are selected to build the method upon.”
> * Neither Phased Value Iteration nor Q-learning converge to an optimal policy with probability 1 in the stochastic (PAC) setting we consider since we always have to account for the sampling failure probability. Classical Phased Value Iteration as well as versions of Q-learning are $\varepsilon$-optimal with probability $1-\delta$ under certain coverage assumptions. However, a key point that is missing in this consideration is that there may be more than one $\varepsilon$-optimal policy. As such, neither of these algorithms are replicable, which is the property we study in this work.
> * We start at the very beginning of sample complexity analysis and use two of the early algorithms that already prove to be quite challenging in the setting we select. Phased Value Iteration (sometimes Phased Q-Learning) - as the name suggests - is an algorithm that effectively does Value Iteration (repeatedly) when having access to sampled transitions rather than the full transition model. This is the first step of introducing stochasticity to dynamic programming-based algorithms such as classical Value Iteration and it already bears a significant overhead for replicability, and necessitates new techniques not previously appearing in the RL literature.
> The second setting we consider is that of episodic exploration and, here, an initial class of algorithms to start with is the $E^3$ or RMax style algorithms. These algorithms contain a natural notion of previously explored states via their sets of known states. We use the sets of known states as a proxy for what parts of the space have been explored and try to replicably have the same sequence of states be added to K. This choice is made to create a way of measuring and comparing exploration across two runs.
>
> “Finite state space and known deterministic reward function seem to be too strong assumptions. The stochasticity comes from 1. exploration and 2. environment stochasticity (including dynamics and reward). If the reward is known and deterministic, a planning algorithm can always be applied and RL is not necessary.”
> * Finite state spaces and deterministic reward functions are very common assumptions in the RL literature and the base problem considering these two has only very recently been considered close to solved (Azar et al., 2017; Zanette and Brunskill, 2019; Simchowitz and Jamieson, 2019). Control problems per se are hard and making these assumptions allows us to study them in parts. As mentioned in the text, our algorithms can easily be adjusted to handle stochastic rewards (see L54f) since estimating them on top of transitions or values is just additive overhead. We agree that planning and RL are connected but both warrant their individual study to make scientific progress.
>
> “With the parallel sampling definition, the parallel sampling subroutine PS(G) actually turns the transition distribution into a deterministic kernel, this is also a strong assumption. To this end, all stochasticity sources are canceled out in your study.”
> * This is incorrect and the parallel sampling subroutine does **not** automatically give us a kernel. This would only be true in the limit of $n$ where $n$ is the number of samples for every individual state. However, sample complexity bounds study the lowest number of samples we can use to achieve our goal which does not require full estimation of the kernel itself (see Kearns et al. 1998). The challenge for the first algorithm is in fact to obtain a replicable policy under the uncertainty of sampling which is not canceled out. If this were true, standard algorithms like Phased Value Iteration would provide replicable policies but they do not (see Figure 1 for counter-example).
>
> “For the experiment section, have the authors tried to experiment on some standard RL environment? Inclusively the classical navigation tasks (but are more standard ones) so that the trade-offs between replicability and the performance can have a clearer exposition?”
> * Our algorithms are tabular RL algorithms and we evaluate them on a tabular MDP in section 5. The experiments we conduct are explicitly designed with two goals in mind: a proof of concept for replicable algorithms in a non-trivial environment (i.e., an environment where there is more than one optimal policy, so that replicability is not already implied by optimality) and to show that the sample complexity overhead of replicability is not as onerous as might be suggested from the asymptotic upper-bounds. Our experiments cover both points.
> Tabular RL algorithms do not generally scale well with respect to the size of the state-space, and so are not typically tested in these environments. Given this, and that our primary contributions are of a theoretical nature, we believe that gym environment experiments would detract from, rather than complement, the presentation of our main results. That said, if there are suggestions for experiments that might improve our understanding of replicability in practice, we would be happy to run them!
> * Could you specify which classical navigation tasks you are referring to? We considered several tasks from the standard RL book by Sutton et al. (2018) but found them too simple for the purposes of studying replicability. A key property for a good benchmark MDP is that there should be several $\varepsilon$-optimal policies that could be selected, so that we can demonstrate our algorithm chooses the same one with high probability over samples (for a fixed random string), and is therefore replicable.

---

> > ### Comment · Reviewer_u4VH · 2023-08-14
> > **Thank you for the response**
> >
> > I appreciate the authors' detailed and thorough responses to my questions, most of them are well-addressed. I still have a few questions remain:
> >
> > 1. It is better to have definitions self-contained. e.g., the definition of sampling failure probability.
> >
> > 2. Can the authors explain what insights can be drawn for the Deep-RL community from the finite space analysis? For example, does it indicates that sampling sufficiently many examples in every state (assuming we are able to query the dynamic models any many times as we need) will improve the consistency of learned policy?
> >
> > 3. On the contributions to the community given the current discoveries: while it is true that reproducibility in RL is essential, it is often in the sense that policies with similar performance can be achieved with high probability. For instance, in your Figure 1, I don't see finding the different approaches in solving the task as a bad property, and the factual problem RL algorithms have is more of they can not converge to similar performance. And this leads to large variance in RL evaluation and comparisons. The intrinsic difficulty is the stochasticity and intricate system dynamics during learning. And a natural approach to me seems to be starting from the offline-RL setting, where datasets are fixed and no stochasticity is introduced by exploration.
> >   I've read the authors' responses to the AC regarding the same issue, and I acknowledge the good properties of such a definition. But as far as I'm concerned, those properties are somewhat driven by the analytical perspective, rather than driven by the existing problems in RL.
> >
> > 4. I understand the time remaining may not be sufficient for experiments, but I wonder if the authors can explain what changes should be made and what difficulty there might be to experiment in, for example, the MiniGrid suite.
> >
> > Many thanks again for your response!

---

> > > ### Author Response · Authors · 2023-08-15
> > >
> > > Thank you for the additional questions!
> > >
> > > **1. - Sampling failure probability**
> > > We apologize for not being more specific here. The sampling failure probability is a common term in theoretical machine learning and refers to the probability that we draw a poor sample that is not representative of the population. For instance, imagine you are trying to estimate the bias of a coin and you flip the coin $10,000$ times. It is possible that all $10,000$ flips come up as heads. While not very probable, this scenario is not impossible. The sample failure probability accounts for these cases where we just got unlucky with our sample. It is commonly denoted $\delta$ and needs to be considered whenever we draw samples from a distribution. For instance, you can find definitions for it in our analysis in both Theorems.
> > >
> > > **2 - Practical insights for the deep RL community**
> > >
> > > A first crucial observation from our work is that replicability might not be computationally impossible to achieve. If we were not able to achieve replicability for RL procedures in finite sized MDPs, one might argue that there is little hope for large state-space MDPs. Our algorithms are a first stab at achieving such replicability and demonstrate viability.
> > >
> > > Second, note that our algorithm does not achieve replicability by simply drawing more samples than the non-replicable versions of the algorithms. As such, we do not think that an insight from our work would be that drawing more samples will improve consistency of the learned policies. Clever randomized rounding and thresholding of the value function are required in order to obtain formal replicability and we believe that these might in fact prove useful tools when developing new deep reinforcement learning algorithms. We are actively considering how to integrate these efficiently into practical deep RL algorithms. As mentioned in our response to the AC, this might show itself as differentiable rounding procedures in offline RL or via meticulously selecting which data points to consider in novel experience replay techniques.

---

> > > > ### Author Response · Authors · 2023-08-15
> > > >
> > > >
> > > > **3 - Importance of the contributions**
> > > >
> > > > We believe there are several challenges the deep RL community is facing that the framework we are proposing can alleviate in the future. These concrete, practical challenges are some of the reasons for developing our replicable algorithms.
> > > > * One big problem is the inability to audit algorithm implementation correctness due to high variance in results from simple deviations in implementation. Replicability gives us a tool for certifying correctness. Whether we take a stable-baselines3 SAC implementation [(https://github.com/DLR-RM/stable-baselines3)](https://github.com/DLR-RM/stable-baselines3) or denisyarats SAC [(https://github.com/denisyarats/pytorch_sac)](https://github.com/denisyarats/pytorch_sac) implementation makes a significant difference on common benchmarks such as the gym MuJoCo tasks.
> > > > * Another problem is that evaluation of algorithms is costly and consequently, reported results are often inconclusive due to minor changes in settings, hyperparameters or even seeds. Offering additional ways to evaluate algorithms with low variance will be beneficial to the field to determine superiority of algorithms, especially when optimizing reward is not the only objective one is interested in (e.g. one might also be interested in developing diverse behaviors).
> > > > * Lastly, we agree that a large part of the challenge in reinforcement learning comes from stochastic and complex system dynamics. Replicability gives us a tool to stabilize learned policies in stochastic settings, allowing us to analyze other moving parts of the problem more easily (such as behavior of agents under differing system dynamics). This is important because often it is not possible to determine which of the many moving pieces in the RL routine is having which effect exactly and they are often very intertwined and convoluted. Formal replicability would allow us to study our algorithms while using several of these fixed pieces as independent variables.
> > > >
> > > > **Offline RL as a starting point**
> > > > This is an excellent point and our present work is in fact starting with a setting that does not exhibit stochasticity from exploration and is very similar to offline RL. One can think of the parallel sampling setting as getting a sequence of (sub)-datasets where we assume that we get data points for every possible state. This setting could similarly be phrased as a batch RL problem where one has access to a single dataset that is fed to the algorithm in pieces. Note that this is somewhat even a step before offline RL because we are assuming that we have data for every state-action pair and conservatism (as a version of exploration) is not a challenge that needs to be considered. For first function-approximation results, we agree that a similar offline version is going to be a very interesting experiment in the future.
> > > >
> > > > **4 - MiniGrid suite experiments**
> > > > Thank you for clarifying which environments this point was referring to. Other than time constraints, nothing would stop us from using some of the mini-grid environments. However, we believe that these environments might not properly strengthen our claims as MiniGrid environments are
> > > > * deterministic and do not have stochastic transitions. We feel that proving replicability in highly stochastic scenarios provides a stronger presentation of the capabilities of our algorithms.
> > > > * explicitly designed to present challenges for deep RL agents. For example, they are often
> > > >     * used to study partial observability which we do not consider in our framework.
> > > >     * used to study planning from image-input which is not considered in our framework (e.g. generalization across different colors of objects).
> > > >
> > > > Specifically the latter points will become significantly more interesting once we have made a leap to linear function approximation which, as we mentioned, we are currently working on for future publication.
> > > >
> > > > If we were to consider MiniGrid environments, one experimental contribution might be to validate the correctness of our exploration algorithm. From a technical perspective, the biggest challenges would be to implement a core of MiniGrid that supports stochastic transitions, write code for the correct sampling wrappers, implement RepRMax and find suitable algorithm parameters.
> > > >
> > > > It is not clear that we could get results before the end of the discussion period but if the reviewer feels strongly that this would improve the manuscript, we are happy to try and include an experiment on MiniGrid in the camera-ready version of our paper should it be accepted.

---

> ### Comment · Reviewer_u4VH · 2023-08-18
> **Raise My Score**
>
> I really appreciate the authors' detailed response. Most of my concerns have been well addressed. So I raised my score from a 4 to 6.
>
> To better enhance clarity, I would like to suggest the authors mention the explicit definition of replicability in policy learning in an earlier stage of the paper, preferably in the abstract/intro, to avoid any potential confusion and inaccurate expectation (e.g., distinguish from the more intuitive definition of "achieving similar performance").

---

### Official Review · Reviewer_fsAq · 2023-07-07

**Soundness:** 3 good
**Presentation:** 2 fair
**Contribution:** 2 fair
**Rating:** 4
**Confidence:** 4

**Summary:**

This paper studies replicable reinforcement learning. And they show that stochastic sample-based value iteration can be done replicably and explore the space of an MDP to find an optimal policy. Furthermore, they give some theoretical results. The effectiveness of the replicable algorithm is validated by simple experiments, requiring fewer sample than the theory suggests.

**Strengths:**

- They first introduce the notion of replicability to RL because RL algorithms are difficult to reproduce.
- They provide two novel algorithms (replicable phased value iteration and Exploration) for replicable RL and also give corresponding theoretical results. This formulation provides a good foundation for the problem.

**Weaknesses:**

- From this paper,  at each iteration $t$, many episodes interacting with the environment are required. This is equivalent to modeling the environment. If so, it is trivial to obtain similar policies, even though there are some theories. The studies in this area concern a few episodes interacting with env at each iteration; after $T$ iterations, the two policies are close to each other. This is just like the Policy Iteration and the Value Iteration.
- The experimental evaluation is weak, though the paper pays more to the theoretical part.

**Questions:**

- Definition 2.3 is not clear. What does $r$ mean? This is the same definition as Impagliazzo et al. (2022). There is no citation!
- ''As a result, we expect that replicable estimation of MDPs is the hardest setting in stochastic RL, followed by replicable value function and then policy estimation'', if we don't know the MDP, how do we estimate the fine value function and policy function. I think that MDP is a foundation.
- ''a new threshold k′ is sampled uniformly from [k, k + w].  '', why is k' chosen at random instead of assigning one? What's your motivation for doing this?
- From the left figure of Fig.2, except for the number of steps, it can be seen the choice of $\rho_{SQ}$ is crucial. We should know how to choose it. Can you clarify why $\rho_{SQ}=\frac{\rho}{|S|}$ has a good result? And if we increase the number of steps, when $\rho_{SQ}=\frac{\rho}{|S|}$, do we still get good results?

**Limitations:**

The authors explained the limitations of their work well. I do not see an obvious negative societal impact.

---

> ### Author Rebuttal · Authors · 2023-08-08
>
> Dear Reviewer fsAq,
>
> Thank you for your feedback. We appreciate that you think that our notion of replicability provides a good foundation for studying reproducibility in RL.
>
> In response to the comments in the Weaknesses and Questions sections:
>
> “at each iteration, many episodes interacting with the environment are required. This is equivalent to modeling the environment. If so, it is trivial to obtain similar policies, even though there are some theories.”
> * Simply interacting with the environment would not give us that we can model it. It is significantly harder to model the full environments than to find a policy (see e.g., Kearns et al 1998). Could you elaborate which theories you are talking about? We are unaware of any theory that would ensure replicability and accuracy of any previous RL procedure.
>
> "The studies in this area concern a few episodes interacting with env at each iteration; after iterations, the two policies are close to each other. This is just like the Policy Iteration and the Value Iteration."
> * This seems to be a key misunderstanding. We are not asking that the two policies are close to each other but we are asking they be *identical* across different stochastic interactions with the environment. In our settings, transitions need to be sampled while algorithms like value iteration assume direct access to the transition function. The stochastic version would be Phased Value Iteration which we use in the paper. There is no guarantee for replicability (or policy similarity) in PVI. See Figure 1 for a counter-example.
>
> “The experimental evaluation is weak, though the paper pays more to the theoretical part.”
> * Our algorithms are tabular RL algorithms and we evaluate them on a tabular MDP in section 5. The experiments we conduct are explicitly designed with two goals in mind: a proof of concept for replicable algorithms in a non-trivial environment (i.e., an environment where there is more than one optimal policy, so that replicability is not already implied by optimality) and to show that the sample complexity overhead of replicability is not as onerous as might be suggested from the asymptotic upper-bounds. Our experiments cover both points. That being said, if there are suggestions for experiments that might improve our understanding of replicability in practice, we would be happy to run them!
>
> “Definition 2.3 is not clear.”
> * r is the shared internal randomness mentioned in the definition and should be stated in the text of the definition. Good catch; we added this in L87. We also added the citation to the definition directly to be more precise.
>
> “if we don't know the MDP, how do we estimate the fine value function and policy function”
> * Most RL algorithms do not assume to *know* the MDP, they merely have access to a sampling machine (e.g., an environment). We do not need to estimate the reward function and transition probabilities to run PVI but we can compute a value function without knowing the exact form of the MDP. This is cheaper in terms of sample complexity since the value function is significantly smaller than the transition function.
>
> ''why is k' chosen at random instead of assigning one?"
> * This is done to achieve replicability. Suppose there are two runs of the algorithm (1) and (2). Our goal is to make updates to the set of known states K identical across runs, to establish downstream replicability. In (2), we would like to add to K exactly all elements that were added to K in (1). Assume there was a fixed threshold k. It could be that in expectation we will see some state-action $(s, a)$ pair enough times that it meets the fixed threshold k. Due to the stochastic nature of exploration, the realized transitions in (1) may cause us to visit $(s, a)$ only (k-1) times, while the realized transitions in (2) cause us to see it (k+1) times. A fixed threshold would mean that we don’t add $(s, a)$ to K in (1), but do add it to K in (2), making replicability difficult to reason about. Yet, we can guarantee that the realized number of visits to a state-action pair concentrates around its mean. By randomizing the threshold k over a large interval, we can ensure with high probability that k does not fall between the two realized values for visitation counts. Hence, the algorithm makes the same decisions about whether $(s, a)$ is added to K in both (1) and (2) (see proof of Theorem 4.2). We have updated the paper to provide this motivation in the algorithmic exposition.
>
> “Can you clarify why has $\rho_{SQ} = \frac{\rho}{|S|}$ a good result? And if we increase the number of steps, when $\rho_{SQ} = \frac{\rho}{|S|}$, do we still get good results?”
> * Informally, the replicable statistical query algorithm uses its random string to partition the [0,1] interval into subintervals, and assigns canonical representatives to each subinterval. It empirically estimates the value of the query using its sample, and then returns the nearest canonical representative to the empirical estimate. For a fixed sample, taking smaller values for $\rho_{SQ}$ improves replicability at the cost of accuracy of query responses, by increasing the width of each subinterval of the partition so that there are fewer partition elements overall. This way, two estimates that are close are likely to be rounded to the same representative, but the possible error induced by this rounding will increase.
> * As long as the sample sizes are sufficiently large and $\rho_{SQ}$ is chosen small enough, we expect that we achieve high replicability. The current sample size of 30 runs was not representative to reflect this so we increased the number of runs we compute identical value functions over to 150. The new plots (appended in PDF) show that with a large sample size, any $\rho_{SQ}$ smaller than $\rho_{SQ} = \rho/|S|$ works well. In all cases, $\rho_{SQ} = 0.2$ results in poor performance arguably because the subintervals are too small such that the likelihood of falling into different bins is too high.

---

### Official Review · Reviewer_BN9y · 2023-07-27

**Soundness:** 2 fair
**Presentation:** 2 fair
**Contribution:** 2 fair
**Rating:** 4
**Confidence:** 3

**Summary:**

This paper proposes a new reinforcement learning algorithm based on the replicability crisis and gives proof for the proposed method, providing a new perspective in this field.

**Strengths:**

1. The idea of replicable reinforcement learning is brand new and may provide a new perspective for reinforcement learning.
2. The proof of the related lemma is sufficient and strict.


**Weaknesses:**

1.	The experiments are insufficient. Though the proposed replicable reinforcement learning requires much fewer samples, the performance of the replicable RL lacks.
2.	More comparable experiments (i.e. DDPG, TD3, A3C, SAC, etc.) should be carried out to show the effectiveness of the proposed method.
3.	It is better to provide some insightful analysis or conclusion from the replicable RL.
4.	The organization may lack cohesion and coherence. Though several definitions and lemma are introduced, the relationship and function of each lemma lack a detailed claim.
5.	The proposed replicable RL should be evaluated in the public benchmark such as gym， MuJoCo, etc.


**Questions:**

Please refer to the Weaknesses.

**Limitations:**

The authors have discussed the limitations in details. Nevertheless, the proposed method should be wildly evaluated in different benchmarks and provide some insights to readers, especially for a new idea.

---

> ### Author Rebuttal · Authors · 2023-08-08
>
> Dear Reviewer BN9y,
>
> Thank you for your feedback, we appreciate that you see the direction of replicable reinforcement learning as a novel perspective.
>
> In response to the comments in the Weaknesses and Questions sections:
>
> “The experiments are insufficient. Though the proposed replicable reinforcement learning requires much fewer samples, the performance of the replicable RL lacks.”
> * Would you be willing to clarify this point? Our proposed algorithms require more samples than the standard versions of these algorithms and it is not clear to us what is intended by “requiring fewer samples”. The algorithms we provide are formally replicable **and** $\varepsilon$-optimal. In all experiments, the learned policies are $\varepsilon$-optimal, so the performance can technically not be better than that. (see Theorem 4.1 and 4.2)
>
> “More comparable experiments (i.e. DDPG, TD3, A3C, SAC, etc.) should be carried out to show the effectiveness of the proposed method.”
> *  We think that comparison to common benchmark deep learning methods is inappropriate for this stage of replicable reinforcement learning research. As we show in our work, designing reinforcement learning algorithms that can learn approximately optimal policies while simultaneously satisfying replicability is a challenging task that requires new technical tools and algorithm design. Our work initiates the study of replicable reinforcement learning, building a technical toolkit for replicable RL toolkit that we hope can be used to obtain more performant algorithms in the future, but our techniques in their current form are not applicable to non-linear function approximation or deep learning. Even linear function approximation would require the development of completely new tools for replicability. We agree that this is an interesting direction for future work, and it is a line of research we plan to pursue, but there are substantial technical leaps to make before we can guarantee replicability of deep learning approaches.
>
> “It is better to provide some insightful analysis or conclusion from the replicable RL.”
> * As we state in the introduction, we provide novel algorithms that are formally replicable. (Line 35 and following) These algorithms are provably correct and replicable with high probability (see Theorem 4.1 and 4.2). The analysis of the algorithms is provided either in the Appendix B.1 and B.2 for Algorithm 1 and in large parts in the main text (L215 - 276) or Appendix B.3 for Algorithm 2. The analysis of the empirical results is done in L298-305. The conclusion is provided in the Conclusion & Future work section. Would you mind clarifying what additional analysis we could add to improve the paper?
>
> “The organization may lack cohesion and coherence. Though several definitions and lemma are introduced, the relationship and function of each lemma lack a detailed claim.”
> * As stated in the text, Definitions 2.1 and 2.2 are used to describe the theoretical settings we consider. Definition 2.3, 2.4 and Theorem 2.1 frame the context of the work and provide the required formal background. Definition 3.1 is one of the contributions and is used to derive the algorithms we propose. Both Theorem 4.1 and 4.2 satisfy this Definition 3.1 (see their respective text). Lemma 4.1 is used to provide the accuracy argument for Theorem 4.1 (see L166-167) (see full proof in Appendix B.1 and B.2). Lemma 4.2 (L215, typo here) is used to prove convergence of Theorem 4.2 (see L230). This should cover all the definitions and lemmas in the main text and all of them are explicitly used in some main claim or are used to frame context and setting. Can you elaborate which specific definition or lemma is unclear?
>
> “The proposed replicable RL should be evaluated in the public benchmark such as gym， MuJoCo, etc.”
> * Our Algorithms are tabular reinforcement learning algorithms and we evaluate our algorithm on a tabular MDP in section 5. The experiments we conduct are explicitly designed to demonstrate that our algorithms replicably converge to good policies, not just that they learn good policies, and moreover that the sample complexity overhead of achieving replicability in practice may be less significant than what is suggested by the asymptotic upper-bounds of our theoretical guarantees.
> * Tabular reinforcement learning algorithms do not generally scale well with respect to the size of the state-space, and so are not typically tested in gym environments. Given that our algorithms are tabular, and that our primary contributions are of a theoretical nature, we believe that gym environment (or similar) experiments would detract from, rather than complement, the presentation of our main results. That being said, if there are suggestions for experiments that might improve our understanding of replicability in practice, we would be happy to run them!

---

> > ### Comment · Reviewer_BN9y · 2023-08-19
> >
> > Thanks for the clarification provided by the authors.
> >
> > The reviwer tend to keep the original score.

---

### Author Rebuttal · Authors · 2023-08-08

Dear Reviewers,

Thank you for your valuable feedback! We are grateful for the questions, suggestions, and opportunity to clarify the intent of our work.

We would first like to provide a set of common answers to the points raised by reviewers here. We will provide the responses to individual reviewer’s questions in the appropriate, reviewer-specific response form.

First and foremost, we would like to highlight that this is a theoretical manuscript with a focus on providing the first-ever results for formally replicable reinforcement learning. Experimentation is secondary and is done to validate the potential usefulness of general replicable algorithms for real-world problems in the future. We are excited that several reviewers see the novelty and need for such algorithms and that the reviewers believe that our claims are well supported. For other commonly highlighted comments or questions:

Deep Learning Baselines: Several reviews ask about comparisons to deep learning methods. We can only stress again that this is the first fundamental work on formally replicable reinforcement learning, and we think that comparison to common benchmark deep learning methods is inappropriate for this stage of replicable reinforcement learning research. As we show in our work, designing reinforcement learning algorithms that can learn approximately optimal policies while simultaneously satisfying replicability is a challenging task that requires new technical tools and algorithm design. Our work initiates the study of replicable reinforcement learning, building a technical toolkit for replicable RL that we hope to further develop in the future, but our techniques in their current form are not applicable to non-linear function approximation or deep learning. Even linear function approximation would require the development of completely new tools for replicability. We agree that this is an interesting direction for future work, and it is a line of research we plan to pursue, but there are several substantial technical leaps to make before we can guarantee replicability of deep learning approaches.

Choice of algorithms. We start at the very beginning of sample complexity analysis and use two of the early algorithms that already prove to be quite challenging in the setting we select. Phased Value Iteration (sometimes Phased Q-Learning) - as the name suggests - is an algorithm that effectively does Value Iteration (repeatedly) when having access to sampled transitions rather than the full transition model. This is the first step of introducing stochasticity to dynamic programming-based algorithms such as classical Value Iteration and it already bears a significant overhead for replicability, and necessitates new techniques not previously appearing in the RL literature.
The second setting we consider is that of episodic exploration and, here, an initial class of algorithms to start with is the E^3 or RMax style algorithms. These algorithms contain a natural notion of previously explored states via their sets of known states. We use these sets of known states as a proxy for what parts of the space have been explored and try to replicably have the same sequence of states be added to K. This choice is made to create a way of measuring and comparing exploration across two runs.

Experiments: Our algorithms are tabular reinforcement learning algorithms and we evaluate our algorithms on a tabular MDP in section 5. The experiments we conduct are explicitly designed with two goals in mind: a proof of concept for replicable algorithms in a non-trivial environment (i.e., an environment where there is more than one optimal policy, so that replicability is not already implied by optimality) and to show that the sample complexity overhead of replicability is not as onerous as might be suggested from the asymptotic upper-bounds. Our experiments cover both points.

It was suggested that we include experiments in a classical gym environment as well. Tabular reinforcement learning algorithms do not generally scale well with respect to the size of the state-space, and so are not typically tested in these environments. Given that our algorithms are tabular, and that our primary contributions are of a theoretical nature, we believe that gym environment (or similar) experiments would detract from, rather than complement, the presentation of our main results. That being said, if there are suggestions for experiments that might improve our understanding of replicability in practice, we would be happy to run them!

We have integrated your feedback into the draft of the manuscript. Here is a brief summary of the changes we are going to add:

1.) We found a minor bug in the analysis of the Replicable R-Max algorithm which we have fixed, and adjusted the rates accordingly.
2.) We made clarifying changes on the notation in Definition 2.3 and fixed a typo in the enumeration of Theorem from (previously) 4.1 to (now) 4.2.
3.) We updated the experimental plots using a more representative sample (new plots in PDF) and included a more intuitive explanation of the analyzed parameter $\rho_{SQ}$.
4.) We extended the informal description of the RepRMAX algorithm to provide better intuition for the randomized threshold.

---

### Comment · Area_Chair_y9Me · 2023-08-08
**Comments from the Area Chair**

I wanted to kick-start the discussion with the authors by providing my perspective on the paper.

The paper addresses a hugely important practical problem in reinforcement learning, namely the replicability of experiments. This has been recognised as a major obstacle to progress and any attempts to address it are very welcome.

Essentially, the authors of the paper are saying: if I sacrifice some sample efficiency (OK, admittedly, it is a lot of sample efficiency), I can guarantee that two (or more) independent runs of an RL algorithm will return exactly the same policy with high probability. The paper lays a foundation for the study of replicability of RL and provides a definitional framework on which future refinements can build on. The authors provide upper bounds for the number of samples required to make variants of phased Q-learning and RMAX replicable. The also try to estimate some lower bounds empirically in their experimental section for a gridworld-style problem.

I have the following main question to the authors. One of the goals of replicability is arguably not as much to have an identical / similar policy on each run, but to have the same / similar performance / return of the agent on each run. Assuming performance / return is bounded, I can always achieve this by running training N times (sacrificing a factor of N in terms of sample efficiency) and reducing the variance in the estimate of mean performance N times (by the central limit theorem). This process is much simpler than your framework and has that benefit that it can be applied to any existing algorithm. I am not arguing that your approach is wrong, but just mostly trying to get you to discuss the benefits of really having exactly the same policy / value function for each seed. This is important given the reward hypothesis underlying most of RL, which implies that policies that achieve the same expected return are equally good.

I also wanted to make some points concerning the applicability of the work to scenarios where deep Reinforcement Learning would typically be used. I do agree with the authors that an empirical comparison with deep learning baselines seems completely out of scope. However, I sympathise to an extent with deep RL practitioners, who want to get practical insights on how to make variants of their algorithms have less variance in performance / more replicability, while still keeping the most important parts of their algorithms the same they have always been using for practical reasons. Therefore I think it might make sense for the authors to add a short paragraph / section that at least tries to answer or scope solutions for the following questions:
- is it possible to have a very large or continuous state space / action space and still enjoy a degree replicability (assuming we have a good inductive bias)
- is it possible to use insights from this work to make variants of deep RL methods that have less variance in performance than the currently used ones (without completely throwing the existing algorithms overboard, but sacrificing the theoretical guarantee on replicability). In a nutshell, is there a practical version of rSTAT I can somehow wrap inside a DQN agent or something like that that will make it have less variance in performance at the cost of decreased sample efficiency?

Of course I understand the questions above might possibly be very hard to answer and require writing a new paper each to answer fully. That is why I am not asking for a full exposition but only for a brief discussion. I also wanted to emphasise that "no" is a perfectly valid answer to the questions above and not a reason to reject the paper.

Minor comments:
- The notation $\tau$ is overloaded too much, being used both for trajectories and as the tolerance/accuracy parameter in rSTAT queries. Please change the notation so that the symbol has one meaning only.
- In line 101, the sample complexity of rSTAT does not depend on the tolerance parameter. I am not an expert on this, but possibly you my have a typo where $\varepsilon$ is used in place of $\tau$. If the number of samples really does not depend on $\tau$ you should provide an intuition why.
- In Definition 3.1, you should clarify whether the event under the probability under line 119 contains an existential quantifier over all state action pairs (i.e. we guarantee that the policy is the same for all action pairs), or whether the whole Definition 3.1 is parameterised by a single concrete state action pair. I think the former is the case but this should be stated explicitly.

---

> ### Author Response · Authors · 2023-08-12
>
> Dear NeurIPS area chair y9Me,
> Thank you very much for taking such a detailed look at our paper; we really appreciate it. We are excited that you agree with us on the importance of the problem of replicability in reinforcement learning.
>
> As response to the first question on the choice of exactly identical policies and the proposed alternative.
> * In general, this intuition and your proposed alternative are correct—when the algorithm optimizes for a single metric, and this is the only metric we care about. However, consider a scenario where one is interested in several properties rather than just reward, e.g., some notion of simplicity. Now, what it means for two policies to be close together becomes much more ambiguous and formal replicability, as we consider, can be helpful to discern the various components of the problem.
> * We agree that the notion of replicability we consider for now is very strong but this strength comes with several upsides. Several of the key benefits of formal replicability as we define it can be summarized as follows.
>
> 1. Formal replicability is auditable. This is in contrast to related stability notions such as differential privacy, which have been shown to be computationally hard to verify (Gilbert,McMillan’18; Gaboardi,Nissim,Purser’20). Previous work has shown that different implementations can quickly lead to greatly differing results in deep reinforcement learning (Henderson et al. 2018). Replicable algorithms are efficiently testable since one can easily choose various different random strings with few different samples to check for correctness of the implementation. This strongly facilitates replication efforts of any algorithm.
> 2. Formal replicability provides a good benchmark. Most RL algorithms do not explicitly consider replicability at all and as such the community has made little progress on the problem. Our notion of replicability provides the strongest benchmark on the other end of the spectrum where replicability is in the center of analysis. It is, so to say, a best-case scenario for replicability. While relaxed notions of replicability might intuitively be closer to what we care about in practice, if they are no more efficient than the strong replicability we are asking for, then we might as well just work with the strong notion as is and use it. A strong notion of replicability allows us to test the limits of what can be achieved when it comes to algorithmic replication.
>
> 3. Formal replicability provides connections to the literature on replicability in supervised settings. Our notion of replicable policies parallels how replicability is explored in supervised as well as unsupervised settings (Impagliazzo et al 2022, Esfandiari et al., 2023). This will allow us to utilize various insights and techniques that have been developed such as connections to privacy and generalization (Bun et al. 2023). It will also allow us to understand how the challenges that RL faces with respect to replicability differ from other common settings and whether RL replicability is strictly harder than common supervised learning replicability.
> 4. Formal replicability facilitates evaluation of methods. In other domains, low variance across runs has facilitated progress and made it easy to establish generally working architectures and methods. Deep reinforcement learning especially suffers from high variance results (Henderson et al., 2018) and in order to obtain reasonable mean estimates as suggested by your alternative approach one might require a large amount of seeds to draw definitive conclusions (Colas et al., 2018). If we can derive algorithms that enforce that two runs produce identical results without paying much overhead, we could potentially establish new ways of evaluating our algorithms that may support more direct hypothesis testing.
> 5. Formal replicability helps with algorithm design. While we agree that the main end goal might not necessarily be to have completely identical policies, replicability as we consider it in our work might provide a very useful tool for building algorithms that have low between-run variance and increasing our understanding of current approaches. One of the big problems in the RL domain is that there are various moving parts and it can be hard to identify which parts are working well. The notion of identical policies and value functions ought to give us a tool to hold part of the problem fixed while we examine other moving parts and allow us to understand where the challenges with current approaches are more clearly. For example, consider two runs of the same algorithm where the policies start disagreeing early on. From that point on, it can become very hard to reason about how the two runs of exploration evolve over time. It becomes unclear why one of the RL runs failed and the other one did not. Building a toolkit for algorithmic techniques that allow us to keep policies synced across runs will facilitate future algorithm development.

---

> > ### Author Response · Authors · 2023-08-12
> >
> > “Is it possible to have a very large or continuous state space / action space and still enjoy a degree replicability (assuming we have a good inductive bias)”
> >
> > * We strongly believe that this is in fact possible and we can even get replicability as strong as defined in our current work. We are actively working on extended versions of our formulation and application to (linear) regression. Similar principles for replicability should apply in the regression setting and we believe that techniques like randomized thresholding will be useful here as well. Access to replicable linear regression procedures would provide a first step towards handling continuous state-spaces and a step towards application to deep reinforcement learning in the future.
> >
> > “Is it possible to use insights from this work to make variants of deep RL methods that have less variance in performance than the currently used ones (without completely throwing the existing algorithms overboard, but sacrificing the theoretical guarantee on replicability)?”
> >
> > * This is an excellent question. While the sub-routines we use might not be directly applicable in a deep learning setting, we strongly believe that several of the insights we generate theoretically might inspire practical algorithmic novelties for improved stability in deep RL. This was one of the major drivers for us to start working on this problem. As we are making progress on the theoretical side towards tools that are efficiently usable in regression settings, there are several empirical directions we are exploring. These are not direct implementations of the techniques we use and as you mention, have no theoretical guarantees, but are inspired by the ideas that make formal replicability work.
> >
> > 1.) A first idea that would directly employ techniques we use in the paper to deep RL would be to formulate mini-batch gradient computation as a statistical query. This would allow for a direct application of rSTAT to guarantee identical gradient computations. However, this might not be the most efficient way to do things in practice and it is not immediately clear that one might not run into issues when computing these queries on GPUs using CUDA. Still, we are actively thinking about more efficient ways to do better than this in the regression setting.
> >
> > 2.) One of the key components of formal replicability is rounding and thresholding procedures, e.g. rounding during the Bellman update operation. One direction we are actively playing around with is differential rounding procedures for value functions in deep RL to reduce overfitting to decimal points in early training. An extreme of this would be to analyze estimation of continuous value functions as a classification problem (note that there are obstacles here since classification in general does not have a locality property). The latter point connects directly to consideration of distributional RL methods as a candidate for more stable learners and might explain some of their empirical success.
> >
> > 3.) An important unanswered question in deep RL is how much impact exploration has on the early convergence to local minima. It seems that in many cases, good exploration is what differentiates working from non-working deep RL procedures. The current method for replicable exploration we propose is not applicable in continuous state-space settings and probably not in most algorithms that do immediate updates to the neural network. However, it begs the question whether appropriate filtering of data points before updating the network is crucial to learning consistently. This is somewhat supported by improved empirical results from various methods on adjusted experience replay. There might in fact be experience replay methods inspired by our heavy hitters-like algorithm that could lead to improved stability.
> > Minor comments:
> >
> > “The notation $\tau$ is overloaded…”
> > * Good catch, thanks for the pointer. We will change the notation. We realized the same is true for the action space and algorithm notation and will adjust those as well.
> >
> > “In line 101, the sample complexity of rSTAT does not depend on the tolerance parameter….possibly you may have a typo where $\varepsilon$ is used in place of $\tau$.…”
> > * That is in fact a typo, $\varepsilon$ should be $\tau$ (or whatever the new notation will be), thanks!
> >
> > “In Definition 3.1, you should clarify whether the event under the probability under line 119 contains an existential quantifier over all state action pairs (i.e. we guarantee that the policy is the same for all action pairs), or whether the whole Definition 3.1 is parameterised by a single concrete state action pair. I think the former is the case but this should be stated explicitly.”
> > * It does in fact hold for every (s, a) and we will highlight this in the definition.

---

### Author Response · Authors · 2023-08-12
**A note on a related paper**

A similar notion of replicability was considered by a separate set of authors in concurrent, independent work, titled “Replicability in Reinforcement Learning”. This work was only released after the original submission deadline for NeurIPS. We will include text along the following lines into the final version of the paper to highlight the differences between the two works.

A concurrent and independent work by Karbasi et al. [2023] also studies formal replicability of reinforcement learning. They also study the setting of discounted tabular MDPs, with access to a generative model, and show the same sample complexity upper-bounds for achieving replicable policy estimation in this setting that we prove in this work (up to polynomial dependence on the discount factor). Additionally, they provide a matching lower bound. They go on to consider two relaxed notions of replicability that allow them to provide better upper-bounds in the generative model setting. Our work instead considers a second setting, providing a first algorithm for replicable policy estimation in the episodic exploration setting. We also provide experimental validation of the practical feasibility of these algorithms.

---

### Comment · Area_Chair_y9Me · 2023-08-18
**Additional Review.**

In order to facilitate a better discussion on the paper, it was decided to request another review. Since the review was requested late in the process (and arrived via email), I am posting it using my AC account. The review has the same validity as the others. The review follows below.

Summary: The author studies a very important and interesting problem in RL, i.e., the replication issue in online RL. As the author mentioned, “The question that arises is which of the many RL objects should be made replicable? We separate the difficulty of replicability into three levels: replicability of the MDP, the value function, and the policy.”



Strengths and Weaknesses:

Originality: I believe this is a novel research problem in the RL context. The study is largely inspired by the counterpart in supervised learning, as the author has pointed out and discussed in detail.

Quality: I don’t get a chance to check the proof since I received the revision task only a few days before DDL, but to me the formulation and presentation of the results are formal, and the result itself seems convincing to me.

Clarity: The paper is mostly clear except the treatment on rSTAT (see limitations).

Significance: The results open up some new research opportunities for the theory community to re-think about the replicability of RL. I believe many other theory researchers should benefit from reading the paper.

Questions: To me, the replicability of the MDP is rather vague (maybe you mean an estimation of the transition?), and the value function is very artificial: For policy-based method you may never use value function. What I can think of is the following four layers of replicability?
1.trajectory level: this is the strongest of course
2.state-action pair level: the empirical distribution of the state-action pair
3.policy level: which is discussed in the paper and I agree for a lot of the cases this is the most important
4.performance level: this is exactly what the provably efficient algorithms paper are studying.

As a result, what the authors study is actually a stronger version of provable guarantees (4). The authors may want to mention this explicitly because this will connect the paper to a lot of other research. Also, I think it’s not hard to prove that 3 implies 4 under certain smooth conditions. This will help the authors place the paper in existing research in RL.

For 1 and 2 they seems less important for the sake of finding a good policy, but they are very natural because these are the objects that you can observe directly and they don’t depend on the algorithm used. It might be an interesting direction to delve into them in the future (knowing whether they are possible in tabular case is already interesting I think!).

Maybe to me an even more basic question is: which of the above are achievable and which are not? This will have an important implication on what we should expect when replicating RL algorithms. And this paper basically give a positive answer for 3.

Of course I haven’t thought about internal randomness, which can be an interesting issue itself. But his is not that important in the interest of the paper, since the two algorithms discussed doesn’t contain internal randomness (unlike Thompson sampling or its variants).



Limitations:

I’m overall satisfied with the writing of the paper but want to point out some minor issues below.

What is rSTAT? Do we only know the existence of such algorithms or actually it can be written explicitly. Also, is it a polynomial time algorithm? Even if the author doesn’t have enough space for it, I guess you want to mention a short answer to the above questions for audience don’t have background about it (like me).

Line 73: I don’t think Episodic setting is a good name here. We can also look into other kinds of settings where strategic exploration is critical, say infinite horizon setting with certain restart. The key is: we on longer have such access to the generator.

Line 101: n ∈ O doesn’t look like an eligible notation to me.



Ethical concerns:

None



Soundness: 4

Presentation: 4

Contribution: 3

Overall: 7

Confidence: 4

---

> ### Author Response · Authors · 2023-08-19
>
> Dear reviewer,
>
> thank you for your insightful comments and the detailed look at our paper. We are excited to see that you share our opinion on the importance of the problem. We are also glad that you believe that our results open up new research opportunities for the community.
> In response to the comments in the Questions and Limitations sections:
>
> **Replicability of MDP**
>
> We consciously made the decision to call it replicability of the MDP because there are an abundance of approaches in the model-based RL community that try to approximate the MDP for planning. Although we don’t actively study this setting, approximation of the MDP is not limited to the transition function but also requires, e.g., approximation of the reward function.
>
> We agree that there are numerous interesting levels of algorithmic behavior for which we might want replicability. The end-goal of the definitions adopted in this work is that the output of the algorithm is replicable. In most cases, the output of the algorithm is either an MDP approximation, a value function estimation, or an estimated policy. Several of the ideas that you mention might in fact play a larger role in obtaining these objects and as you point out both our algorithms are on the stronger side when it comes to guarantees. However, the definition of, e.g. state-action pair replicability, is not quite as well defined and depends on the setting. One might interpret our set of known states $K$ as a proxy for this but not every algorithm requires maintaining such a set. As such, we believe that our definitions are most general because they effectively cover the majority of objects that an RL algorithm would output.
>
> On achievability: This strictly depends on the exact definitions of the replicable objects studied. E.g., if our goal was to obtain identical sequences of trajectories in an exploration setting, that would likely not be achievable due to transition dynamics. Imagine an MDP with one starting state that branches off into two sub-MDPs. Now, suppose there is only one action in the starting state and this action throws us into either sub-MDP with 50% chance. There is no possible way to get strong replicability of trajectories the way I defined it here. That said, we agree that it is an interesting question to understand what assumptions, e.g. on the MDP, are strictly necessary for various notions of replicability to be achievable in different RL settings, and hope to answer some of these questions in the future.
>
> **What is rSTAT?**
>
> This is a very good point and we are going to highlight this more in the paper. The rSTAT algorithm was introduced in ILPS22 and underlies the proof of Theorem 2.1 that we cite. At a very high level, rSTAT uses its sample to empirically estimate the expected value of the statistical query on the target distribution. It then uses its internal randomness to pick an evenly-spaced set of canonical representatives from the [0,1], and returns whichever canonical representative is closest to the empirical estimate. Its sample complexity and runtime are both polynomial in the (inverse of the) replicability, tolerance, and sample failure parameters. We will add an intuitive explanation of this to the paper.

---

### Decision · Program_Chairs · 2023-09-21

**Decision:**

Accept (poster)

**Comment:**

This paper breaks new ground by defining a framework for provably replicable RL algorithms for tabular RL settings and as such is highly valuable for the NeurIPS community. At the cost of reduced sample efficiency, we are guaranteed, with high probability, to always obtain the same policy from the algorithm regardless of the random seed used. Some reviewers have raised concerns about applicability to deep RL settings, but I think these concerns are only founded to a limited extent.  I think the authors have addressed the concerns of the deep RL community adequately in their answer to my comment. In a nutshell, I do not think it is fair to expect a paper based on a premise this novel to match performance of more standard algorithms which have been tuned over many years.

However, the paper still does have some presentation issues. For example:
- $\mathcal{A}$ is used for both action set (sec 2.1) and algorithm (sec 2.2)
-  $r$ is used for both reward (sec 2.1) and algorith randomness (sec 2.2) which is not defined until def 3.1

I wanted to ask the authors to carefully-proof read the paper again to make sure issues like the above are addressed.